# Is There Enough Evidence to Support the Role of Glycosaminoglycans and Proteoglycans in Thoracic Aortic Aneurysm and Dissection?—A Systematic Review

**DOI:** 10.3390/ijms23169200

**Published:** 2022-08-16

**Authors:** Pratik Rai, Lucy Robinson, Hannah A. Davies, Riaz Akhtar, Mark Field, Jillian Madine

**Affiliations:** 1Department of Biochemistry and Systems Biology, Institute of Systems, Molecular and Integrative Biology, Faculty of Health and Life Sciences, University of Liverpool, Liverpool L69 7ZB, UK; 2Department of Mechanical, Materials and Aerospace Engineering, School of Engineering, University of Liverpool, Liverpool L69 3GH, UK; 3Liverpool Centre for Cardiovascular Science, Liverpool L7 8TX, UK; 4Department of Cardiac Surgery, Liverpool Heart and Chest Hospital, Liverpool L14 3PE, UK

**Keywords:** thoracic aortic aneurysm, thoracic aortic dissection, proteoglycans, glycosaminoglycans, biomechanics, aorta

## Abstract

Altered proteoglycan (PG) and glycosaminoglycan (GAG) distribution within the aortic wall has been implicated in thoracic aortic aneurysm and dissection (TAAD). This review was conducted to identify literature reporting the presence, distribution and role of PGs and GAGs in the normal aorta and differences associated with sporadic TAAD to address the question; is there enough evidence to establish the role of GAGs/PGs in TAAD? 75 studies were included, divided into normal aorta (*n* = 51) and TAAD (*n* = 24). There is contradictory data regarding changes in GAGs upon ageing; most studies reported an increase in GAG sub-types, often followed by a decrease upon further ageing. Fourteen studies reported changes in PG/GAG or associated degradation enzyme levels in TAAD, with most increased in disease tissue or serum. We conclude that despite being present at relatively low abundance in the aortic wall, PGs and GAGs play an important role in extracellular matrix maintenance, with differences observed upon ageing and in association with TAAD. However, there is currently insufficient information to establish a cause-effect relationship with an underlying mechanistic understanding of these changes requiring further investigation. Increased PG presence in serum associated with aortic disease highlights the future potential of these biomolecules as diagnostic or prognostic biomarkers.

## 1. Introduction

Glycosaminoglycans (GAGs) are linear chains of repeating disaccharide units. The covalent addition of GAGs to a protein core produces proteoglycans (PG). There are four main classes of GAGs that are implicated in the normal structural and physiological functioning of the human aorta: hyaluronic acid (HA), chondroitin sulphate (CS), heparin sulphate (HS) and keratan sulphate (KS). The aortic vascular smooth muscle also produces diverse classes of PGs, which can be characterised according to their GAG component and identified within the vasculature: CS-PG Versican and Aggrecan; CS/DS-PG Biglycan and Decorin; HS-PG Perlecan and Syndecan; HS and/or CS-PG Agrin and testican2 and KS-PG Mimecan [1,2,3,4,5,6]. The aorta is the largest blood vessel in the human body, with the responsibility of transporting oxygenated blood from the heart into the systemic circulation. As the thoracic aorta receives blood directly from the heart, it has developed a robust structure to withstand high pressures [7]. The aorta is comprised of three distinct layers; the intima, media and adventitia (Figure 1a,b). The intricate composition of the large medial layer is key to maintaining normal structure and function [8]. It is rich in elastin fibres, which grant elasticity; smooth muscle for contractility and collagen fibres for structural integrity. GAGs and PGs are also present in the thoracic media. These negatively charged molecules are thought to contribute to the maintenance of interlamellar pressure by sequestering water, thus retaining tension in the thoracic aorta [9]. Additionally, PGs are believed to regulate various cytokine, chemokine and protease activities within the extracellular matrix (ECM) [10].

GAGs and PGs have also gained significant interest for their role in aortic pathology, particularly in thoracic aortic aneurysm and dissection (TAAD). Being exposed to a large volume of pulsatile blood flow, the aorta is highly susceptible to mechanical trauma. Over time, segments of the thoracic aortic wall can weaken and dilate. When a segment expansion exceeds 50% of its baseline diameter, a diagnosis of thoracic aortic aneurysm (TAA) is made (Figure 1b,c) [11]. Like any aneurysm, TAAs are at risk of rupture, although the most common complication is thoracic aortic dissection (TAD). Aortic dissection occurs due to the development of a tear in the intima, allowing blood to pool and further propagate the separation of the arterial layers (Figure 1d).

In TAAD, numerous works of literature have reported GAG pooling in the medial layer that is surrounded by localised fragmentation of elastin and collagen fibres, coupled with the scarcity of smooth muscle cells (SMCs) [12,13]. Although the association between GAG/PGs and TAAD has been demonstrated, the cause-and-effect relationship is yet to be established. One popular theory behind the pathological mechanism of GAG pools was proposed by Humphrey et al., where localised GAG pools are hypothesised to increase the local intra-lamellar pressure and disrupt the elastic lamina within the aortic wall [14]. Additionally, the pools are also under consideration as potential sites of initial intimal delamination in TAD due to localised reduction in tensile strength combined with stress concentration generation [15]. However, there is literature which has reported contrasting findings of low GAG levels, with the depletion of GAGs being correlated with reduced viscoelastic properties of the aorta [5]. Currently, GAGs/PGs have been recognised as key macromolecules contributing to the pathogenesis of TAAD, but is there enough evidence to establish the role of GAGs/PGs in TAAD? This review was carried out to probe what is known about the presence and distribution of PGs and GAGs in the normal aorta and in TAAD.

## 2. Methods

A systematic review of literature on the role of GAGs or PGs in human aortas and sporadic TAAD was conducted and reported as per the Preferred Reporting Items for Systematic Reviews and MetaAnalyses (PRISMA) guidelines. Searches were performed, including articles published until 31 December 2021.

### 2.1. Search Strategy and Selection Criteria

The Medical Subject Headings (MeSH) and Boolean terms (glycosaminoglycan OR proteoglycan) AND aort* were searched across four databases: PubMed, Scopus, Web of Science and JSTOR. Two independent reviewers were assigned for title, abstract and full-text review according to the predetermined inclusion/exclusion criteria (Table 1). A third reviewer was consulted if there were any discrepancies and to minimise meta-bias during the process.

### 2.2. Data Extraction and Synthesis

Data from the selected articles for the final analysis were extracted and collated into an excel spreadsheet. Disagreements were resolved through consensus with the third reviewer. Standard data from the articles that included title, year, author, country and study design were tabulated. Alongside the specific data for analysis was tabulated, including what GAGs/PGs were reported, the disease being processed, sample size, age at tissue collection, total GAGs, subtypes of GAGs/PGs reported, tissue location (root, ascending, arch, descending), aneurysm location, whether or not aneurysm occurred, dissection location, whether or not dissection occurred, aortic layers used for GAG extraction and GAGs/PGs extraction and analysis methods.

## 3. Results

From the 8280 studies identified from the search, 75 papers were selected for data extraction after a full-text review (Figure 2) utilising the inclusion/exclusion criteria outlined in Table 1. These were divided into studies on normal human aorta (*n* = 51) and aorta associated with TAAD (*n* = 24). It was noted that earlier studies (1960–1990) mainly focused on optimising methods and identifying GAGs/PGs in normal rather than diseased tissue (Figure 3). Disease studies were largely carried out more recently (1990 onwards) using previously optimised methods with improvements in quantification methods, e.g., immunofluorescence.

### 3.1. Normal Thoracic Aorta

From the 8280 studies identified from the search, 51 papers were selected for data extraction after a full-text review, including information on GAGs/PGs in normal aorta (48 primary laboratory research, 1 computational modelling, 1 short communication, 1 pilot study). Twenty-seven of these studies reported analysis of the composition or ratio of specific GAG subtypes. CS is consistently the highest subtype of GAG identified reported to be present at 40–62.5%, HS is the next abundant GAG with a reported presence of 18–30.5%, followed by DS (12–21.3%) and HA (4.8–15%) [16,17,18,19,20], with KS only reported in 2 studies with 4–8% presence [16,21].

8 studies contained information about the distribution of GAGs in different layers of the aortic wall. Higher total GAG levels have been reported in the intimal layer compared with the outer layers of the aorta [21,22,23,24,25]. Reports on layer-specific distribution of individual GAG subtypes are less consistent, with one study reporting increased DS/HS in the outer layer compared with inner layers [26], consistent with a study reporting DS as the main GAG in the adventitia compared with CS as the main GAGs in the intimal and medial layers [23]. In contrast, another study reported higher DS/HS in the intima compared with the media [22]. A further study found more CS in the inner half of the aortic wall, with DS distributed diffusely throughout the wall, while HA was abundant in the media attached to elastic fibres, with HS found largely in the inner half of the aortic wall [25].

#### 3.1.1. Age-Related Changes in Glycosaminoglycans in Normal Aorta

From the search, 13 papers reported changes in GAGs with age without association with disease. An additional TAAD paper also reported age-related changes in normal tissue [27]. The majority of studies report total GAGs to increase with age, accompanied by an alteration of ratios of subtypes (Table 2). These include an increase in ratio of sulphated:non-sulphated GAGs [28], increase in CS [27,29], particularly Chondroitin-6-sulphate (C6S) [23,30], CS/DS [31] and an increase in CS:HA ratio [32]. An increase in sulphation of HS and increase in the degree of protein-GAG cross-linking has also been reported upon ageing [33]. One study noted an increase in non-uniform distribution of GAGs across the layers upon ageing, with GAGs found in all three layers of the old aorta studied (with adventitial levels less than the other layers), compared with an absence of GAGs in the adventitia of the young aorta [24].

There was only one study that contradicted these findings and reported a decrease in total GAG content with age [16]. This study was also the only study to investigate GAG concentrations across the length of the aorta with ‘young’ (31–38 years) showing an increase in GAG concentration from proximal to distal regions, compared with a consistent level observed for older people (69–76 years) along the length of the aorta. This leads to different trends associated with ageing for specific GAG sub-types dependent on the area of the aorta being studied (Table 2).

In addition to most studies reporting overall increases in total GAG levels upon ageing, some studies have reported initial increases followed by a decrease with further ageing. Kumar et al. and Schorah et al. report an increase in total GAGs up to age 30–40, followed by a decrease [20,34]. These studies also noted more changes in composition in the intima than the medial layer, with DS and C6S increasing in concentration, then decreasing, C4S progressively increasing, while HA and HS reached peak levels at an early age and then gradually decreased. Bertelsen also reported differences between different layers of the aorta; with an increase in hexosamine content observed with age in the media, whereas the intimal layer shows an increase up to 50 years, with a decrease observed upon further ageing [28]. Okado et al. reported similar findings of increased GAG levels during early life, with later decrease [25]. This study divided their analysis by gender and noted that the peak for GAG levels occurred approximately 10 years earlier for males than females (30–40 years, compared with 40–50 years, respectively). They also noted a similar trend for all segments of the aorta from proximal to distal. It was, however, noted that females had higher GAG levels in the proximal and abdominal regions compared with males, whereas levels were comparable between the sexes in the upper descending thoracic aorta.

**Table 2 ijms-23-09200-t002:** **Studies reporting age-associated changes in GAG levels.** Reported increases (↑) and decreases (↓) are shown for total GAG and subtypes: chondroitin-4-sulphate (C4S), chondroitin-6-sulphate (C6S), dermatan sulphate (DS), heparan sulphate (HS), hyaluronic acid (HA) and keratan sulphate (KS). Methods used for analysis: viscometry (V), electrophoresis (E), chromatography (C) or histology (H) are shown. The region of the aorta used for analysis (ascending or descending) and the age range included in the study are also indicated.

Study	Methods	Total GAG	CS		DS	HS	HA	KS	Region	Age Range (Years)
			C4S	C6S						
[32]	V,E,C		↑				↓			1–84
[29]	E,C		↑			↑ to middle age then ↓	↓			28 week foetus–97
[34]	C	↑ to 30 then ↓		↑ then ↓	↑ then ↓	↑ to 20 then ↓	↓		Descend	1–70 total, 2–42 sub-types
[35]	E,C	↑							Ascend	3–78
[28]	E,C	↑	↑ sulphated:non-sulphated ratio				↓			0 to >71
[20]	C,H	↑ to 40 then ↓		↑ to 70 then ↓	↑ to 40 then ↓	↑ to 20 then ↓	↓		Descend	0–58
[33]	E,C					↑ sulphation			Ascend	Young (20 ± 5), Old (80 ± 5)
[25]	H	↑ then ↓			↑		↓		Whole	4 months–96
[36]	E,C	↑			↑		↓		Descend	Young (1–5) Old (60+)
[23]	E			↑						Young (4 months–5) Old (16–78)
[16]	E,C	↓		↓		↓		↑	Whole	13–76
		↑	↓	↑	↓	↑	↓	↑	Arch	Young (31–38) Old (69–76)
		↓	↑					↑	Upper descend	
		↓	↓	↓	↑	↓	↑	↑	Lower descend	
[31]	C	↑	↑	↑	↑	↓				28–83
[27]	E,H		↑						Ascend/descend	35–84
[30]	E,C	↑	↑	↑						0–82
[24]	H	↑							Descend	14 and 69

One study included an analysis of aortic tissue from newborns, with interesting results; newborn and older adults (70 years+) showed similar contents of GAGs, whereas small children (1–5 years) had significantly less GAG in their aorta [36]. Additionally, one study included foetal GAG measurements, which showed that HS levels were very low during intrauterine life, with a rapid increase towards birth [29].

#### 3.1.2. Proteoglycans in Normal Human Aorta

PGs have been identified in the normal human aorta with associated co-localisation and function; DS-PGs forming a lattice of interconnecting collagen fibrils, HS-PGs associated with cells and interstitial CS-PGs [8,37,38]. Hyaluronan and versican have been identified as aggregates in the medial layer of large arteries [39]. The only reported study investigating the association of PGs with age in the aorta identified in this search reported a change in the glycosylation pattern of aggrecan, resulting in loss of CS binding domains with increasing age [3].

### 3.2. Thoracic Aortic Aneurysm and Dissection

From the 8280 studies identified by the search, 24 papers reported GAGs and PGs in association with TAAD that were isolated for data extraction following full-text review. The two main study designs were primary laboratory research (*n* = 22) and computational modelling (*n* = 2). There were 14 papers that compared RNA or protein levels of GAG/PGs or PG degrading enzymes between human control vs. TAA or TAD (Table 3).

An increase in total GAGs was observed in TAD [40]. CS and HS were increased across the whole aortic wall, whereas HA and DS were found localised to the area of dissection, associated with altered collagen and elastin organisation. This suggests that HA and DS may play a role in local alterations within the aortic wall and, in turn, influence altered mechanical properties around the site of dissection. In contrast, another study found a decrease in sulphated GAG content in TAD compared with control [27]; CS was similar between the groups, whereas HS and DS were decreased. A further study did not identify differences in overall GAG level between control and TAA; however, they did observe differences in the proportion of different sub-types, with increased C6S identified in TAA [41].

An increase in aggrecan, versican, osteoglycin/mimecan and testican2 has been observed in TAAD tissue compared to control [6,42,43]. An increase in RNA levels of versican but not biglycan was also observed in aneurysm patients compared to control [44]. Serum lumican levels were significantly increased in TAD relative to control [45]. Aggrecan and osteglycin/mimecan are also increased in TAD serum, whereas levels of aggrecan in TAA serum were similar to controls [44]. Conversely, a decrease in decorin mRNA has been reported in TAA/TAD, with a corresponding decrease in protein level also reported in TAD [46] compared to no effect on protein levels reported for TAA [6]. One study investigated the presence of hyaluronan, versican, decorin and biglycan in acute dissection, chronic dissection and control patients [47]. Decorin was only observed in the adventitia, whereas the other PGs were located in all layers of the aorta. No differences were observed between control and dissection patients; however, hyaluronan was observed to be present near the dissection tear, with authors suggesting a role in wound healing. Lumican was also identified in intimal and medial layers in TAD patient tissue [48].

#### Proteoglycan Degradation Associated with TAAD

ADAMTS (A disintegrin and metalloproteinase with thrombospondin) are a family of multidomain extracellular protease enzymes that play a role in degrading PGs [49]. Although ADAMTS are a group of enzymes and not GAG/PGs, they are of interest within this study due to their role in PG degradation that may influence aortic tissue structure. Four studies reported changes in ADAMTS levels in TAAD compared with control (Table 3). An increase in ADAMTS-1 and ADAMTS-4 was observed in TAA/TAD tissue and blood samples compared with control [50,51,52]. In a separate study, analysis of mRNA levels of ADAMTS-1, ADAMTS-4, ADAMTS-9, ADAMTS-15 and ADAMTS-20 were not altered between control and TAAD, in contrast to ADAMTS-5 which was decreased in TAAD [42].

Exoglycosidases are glycoside hydrolase enzymes which cleave the glycosidic bond at the terminal residue in a saccharide. One study reported an increase in exoglycosidases activity in aneurysmal compared with normal aortas (Table 3), which may suggest a role of exoglycosidases in altering the concentration or structure of PGs in TAA [53].

**Table 3 ijms-23-09200-t003:** **Studies where a difference was observed between normal and TAAD human diseased tissue.** Each study is included with the GAG/PG/Enzyme presented comparing the human control groups vs. thoracic aortic aneurysm (TAA), thoracic aortic dissection (TAD) or thoracic aneurysm and dissection group (TAAD). Red indicates an observed decrease and green an observed increase in the level of GAG/PG/Enzyme compared with control.

Study	GAG/PG/Enzyme Studied	TAA	TAD	TAAD
[42]	Aggrecan			
[43]	Aggrecan			
[42]	Versican			
[43]	Osteoglycin/mimecan			
[44]	Versican			
[6]	Testican2			
[46]	Decorin			
[6]	Decorin			
[45]	Lumican			
[40]	Total GAG			
[27]	Sulphated GAG			
[41]	C6S			
[42]	ADAMTS-5			
[50]	ADAMTS-4			
[51]	ADAMTS-4			
[51]	ADAMTS-1			
[52]	ADAMTS-1			
[53]	Exoglycosidases			

## 4. Discussion

### 4.1. Functional Role of Glycosaminoglycans and Proteoglycans in Thoracic Aorta

The presence of GAGs/PGs within the media of a healthy thoracic aorta is widely accepted. Although GAGs have a relatively low prevalence of <4% within the thoracic aortic wall, they are reported to have a significant supportive role in the structural and biomechanical properties [10]. It is hypothesised that the negatively charged GAGs sequester water into the medial layer to assist with maintaining tension within the elastic fibres [9]. Consistent with this theory, removal of GAGs from the thoracic aorta results in observed differences in the organisation of surrounding elastin and collagen fibres and altered mechanical properties [5]. In addition to sequestering water, negatively charged GAG chains also attract other positively charged molecules, including growth factors, cytokines and chemokines within the ECM. These interactions form part of the mechanism maintaining local distribution, degradation and orientation essential for controlling cell adhesion and interactions with cell-surface receptors that mediate signalling pathways.

Inhibition of PG synthesis in vitro using cell models highlights the role of PGs in the structural assembly of the aortic ECM and in cell-ECM interactions [54]. PGs are thought to function as multivalent cross-linkers of matrix components, with different PG subtypes varying in their functional role. DS-PGs aid collagen fibrillogenesis, HS-PGs act as a receptor for growth factors and cytokines [8] and CS-PGs contribute to sustaining compression against pulsatile forces [55]. The interplay between the different classes of PGs is essential in maintaining the homeostasis of the aortic wall. Changes in PG expression associated with maintaining ECM homeostasis within the aortic wall have been identified that favour aneurysm or dissection formation through distorting elastin and collagen arrangements or altering proteolysis of ECM components [16].

### 4.2. Age-Related Changes in Glycosaminoglycans and Proteoglycans

The composition of the thoracic media is intricate, demonstrating quantitative variation of the biomolecules across ages. Elastin levels display an inverse relationship, decreasing with increasing age [56]. Meanwhile, collagen levels rise dramatically after 40–50 years of age, accompanied by an increase in the irregular arrangement of the fibres [57]. This literature search has highlighted that there is some inconsistency regarding age-related changes in GAG levels in the non-diseased thoracic aorta. Most studies report an increase in total GAG levels with ageing. Some studies also report an increase in specific subtypes in early age, often with a subsequent decrease upon further ageing. Functional changes may also occur upon ageing. An increase in sulphation of HS and increase in the degree of protein-GAG cross-linking has been reported in the thoracic aorta upon ageing [33]. Functional changes have been probed further using abdominal aortic tissue, which also undergoes an increase in sulphation upon ageing [58]. These alterations have been shown to impact the signalling function of HS by enhancing its binding to platelet-derived growth factors (PDGF) A and B. PDGFs are thought to contribute to smooth muscle cell migration and proliferation associated with pathological changes, including arterial wall enlargement in atherosclerosis [59]. Schorah et al. reported an increase in intimal and medial thickness with age which is associated with increased cell proliferation during early life, with further increases in later life due to fibrosis [20]. The timing of GAG concentration peaks and increase in sulphation of HS upon ageing, therefore, suggests that GAG changes could be associated with increased cell proliferation.

Given the aorta increases in stiffness with age [60], the reported change in the glycosylation pattern of aggrecan, resulting in loss of CS binding domains with increasing age, suggests a role of aggrecan in the biomechanical properties of the aorta [3]. This is consistent with computational modelling suggesting the role of PGs in generating localised stress concentrations in the artery wall [39]. Additionally, increased plasma mimecan levels have been associated with increased arterial stiffness, possibly related to its role in maintaining endothelial and vascular smooth muscle cell integrity [61].

### 4.3. Glycosaminoglycans and Proteoglycans in Thoracic Aortic Aneurysm and Dissection

TAAD demonstrates generalised disruption in the microenvironment of the aortic wall [12], including elastin and collagen fragmentation with SMC loss surrounding areas of GAG pooling [13]. Computational modelling has been used to probe the hypothesis that GAG/PG pooling could lead to the initiation of dissection. These studies suggest that multiple structural disruptions occur as a result of increased swelling pressures generated by localised GAG accumulation. The increased pressure is believed to disrupt the connections between the elastic fibres, SMC and collagen in the matrix, forming an unstable region with the potential to delaminate and dissect [9]. This may form part of a positive feedback mechanism whereby the localised increase in swelling pressures caused by GAG pools can be transferred to surrounding lamellar units and propagate the disruption [62]. Collectively these studies suggest that GAG pools and thus increased interlamellar swelling pressures can advance existing TAAD but may also initiate dissection events by causing delamination of the elastic lamina [14,15].

Despite several studies reporting GAG accumulation, the quantification of the total and subtype of GAGs in TAAD tissue is limited (see Table 3). An overall increase in total GAG levels compared with control tissue has been reported. The relative prevalence of GAG subtypes is the same between normal and diseased aorta, with CS being the highest, followed by HS, DS and then HA [40]. Differences in distribution between control and TAAD tissue have been observed, with CS and HS increasing diffusely across the aorta, whereas HA and DS were found to have high local concentrations at sites of dissection associated with an abundance of collagen and elastin. Among the GAG subtypes, HA is known for its capacity to bind and retain water molecules [63]. Therefore, HA could be an important component in the GAG pools, contributing greatly to the generation of swelling pressures that eventually delaminate the intima and weaken subcellular connections with other molecules, including collagen and elastin. However, further conflict exists as Gutierrez et al. reported a decrease in HS and DS between TAD and age-matched controls [27]. A decrease in sulphated GAG content in TAD compared with control could be consistent with altered SMC proliferation in TAD [27].

Aggrecan and versican have been reported to be increased in TAAD (Table 3) and can form aggregates within the aortic wall. Aggregates may contribute to the disruption of local tissue architecture observed in TAAD tissue [42]. In addition to CS chains, aggrecan and versican also contain HA binding domains. HA has been associated with controlling pressure, being observed near the dissection tear suggesting a role in wound healing; however, this was not consistent across all samples studied [47]. In contrast, abdominal aortic aneurysms (AAA) appear to have decreased versican concentration and mRNA levels [64]. Fragmentation and structural modifications of PGs (specifically versican) have also been reported in AAA [65,66]. These studies point towards different roles for PGs/GAGs in TAA and AAA and may reflect additional differences in aortic structure across the aortic tree. One suggested a role for PG association with aortic disease is that versican levels increase in the early stages of disease associated with early remodelling of the ECM, with subsequent degradation of versican occurring following upregulation of degradation enzymes [10]. Versican and HA have been proposed to produce a pro-inflammatory response within the ECM, including stimulation of cytokine release (Figure 4) [67].

Decorin has been proposed to have multiple roles depending on where it is located within the aortic wall, which may influence the development of aortic aneurysms. Decorin within the adventitia has been suggested to protect against aneurysm formation, with decreased levels observed in AAA, whereas when present within macrophages, it is thought to promote aneurysm formation within the abdominal aorta [68]. Together with biglycan, decorin is known to promote collagen fibrinogenesis, in turn contributing to the regulation of aortic wall integrity. Consistent with these roles’ the expression of recombinant decorin in mice rescues Ang II-induced AAA formation and rupture. Despite the limited literature available regarding the role of decorin in TAAD, a decrease has been observed in TAA and TAD (Table 3) [6,46]. Coupled with observations that it is only found within the adventitia [47] and AAA studies suggesting decorin may play a critical protective role in ECM matrix maintenance, decorin may, therefore, represent an avenue for future exploration and therapeutic targeting.

Despite no difference in biglycan gene expression being observed in non-syndromic TAA [44], a loss-of-function mutation in the biglycan gene is associated with early-onset TAAD, coupled with increased TGF-β signalling [69]. Additionally, biglycan-deficient mice develop dissection and suffer aortic rupture [70]. Examination of the aortas of biglycan deficient mice revealed structural alterations of collagen fibrils with impaired tensile strength. This implies that biglycan plays a structural and functional role in maintaining aortic wall integrity.

### 4.4. Animal Studies in TAAD

There have been many animal studies that focus on ADAMTS deficiency to investigate the effect on PGs in TAAD (Table 4). Firstly, a murine model with ADAMTS-5 deficiency used to investigate aggrecan cleavage during aortic wall development found that mice deficient in ADAMTS-5 exhibit ascending aortic wall defects and altered proteolytic cleavage of aggrecan [71]. A similar study using ADAMTS-5 deficient mice found versican was the most upregulated ECM protein and identified ADAMTS-1 as a key protease for versican regulation in murine aortas [72]. These studies were consistent with observations from a murine model of severe Marfan syndrome (a syndromic genetic connective tissue disorder), with increased aggrecan and versican mRNA and reduced expression of ADAMTS-5 [42]. It has also been suggested that ADAMTS-9 may have a role in aortic anomalies. Mice lacking ADAMTS-9 were associated with accumulation of versican whilst mice in the control wild-type litter showed a decrease in cleaved versican [73]. Ren et al. utilised a murine model to probe the involvement of ADAMTS-4 and confirm their findings that ADAMTS-4 levels were increased in human TAAD samples (Table 3) [50]. They showed that ADAMTS-4 deficiency resulted in reduced aneurysm and dissection formation, with associated maintenance of normal elastin fibre arrangements, reduced inflammatory cell infiltration and apoptosis, with reduced versican degradation. These animal studies support the role of ADAMTS proteases and associated PGs in TAAD.

Many TAAD animal studies use an angiotensin II (AngII) induced model, in which mice are infused with AngII, a method commonly used to generate TAD. ADAMTS-1 was found to be overexpressed in macrophages and neutrophils infiltrating the media of the aorta in these animals [52]. Consistent with enhanced ADAMTS-1 expression, increased versican degradation was observed, suggesting a role of ADAMTS-1 and associated versican degradation in TAD.

AngII can also be combined with β-aminopropionitrile (BAPN) to induce TAD. Lumican double knockout mice treated with BAPN and AngII displayed a significant increase in aortic rupture and AD-associated mortality compared with wild-type BAPN-AngII treated animals [48]. This suggests that lumican may play a protective role in maintaining aortic wall integrity. Additionally, lumican levels were higher in BAPN-AngII wild-type mice compared with non-treated, consistent with increased lumican observed in human TAD serum [45].

Another study discovered biglycan deficiency results in disruption to collagen fibres and up-regulation of vascular perlecan content with an associated increase in aneurysm development in mice [74]. These findings were consistent with biglycan-deficient mice presenting structural abnormalities of collagen fibrils and reduced tensile strength [70]. Both studies implicate biglycan as a key factor in structure and functional integrity within the aortic wall, with a potential role in the pathogenesis of aortic pathologies. Another AngII-infused murine study identified remodelling of the aortic wall following an initial injury involves the synthesis of new collagen, which is co-localised with GAG deposition [75]. This study suggests that GAG deposition could be part of a protective mechanism within the aortic wall against further dissection.

Consistent with literature suggesting a decrease in decorin in TAA and TAD (Table 3), an animal model was investigating Sirtuin 1 (SIRT1) in TAD identified control of decorin expression and associated downstream signally pathways as a potentially protective mechanism [46]. Pools of GAGs consistent with those observed in human aortic tissue have also been observed in murine models of contractile protein mutations associated with altered wall integrity [76]. Perlecan-deficient mice have been proposed as a model for AD showing a high frequency (15–35%) of dissection associated with immature elastic fibres and torn, thin elastic lamina within the aortic wall [77].

### 4.5. Why Are We Interested in GAGs/PGs?

It is becoming evident, as highlighted in this review, that GAGs/PGs are likely associated with TAAD, but why is this important? Any factor identified to be associated with disease initiation and progression provides an opportunity for improving diagnostic capabilities, assessing the risk of rupture and future therapeutic targeting. As described here, lumican and aggrecan have elevated serum levels in aortic disease compared with controls [43,45]. Serum lumican levels have also been reported to be significantly higher in acute compared to chronic AD patients [48] and correlated with risks of post-operative complications following aortic surgery [78]. While further work is needed to fully understand the relationship between GAG/PG levels, stage of disease and prognosis, this data is promising to suggest that GAGs/PGs could have potential as blood biomarkers for aortic disease diagnosis and risk/prognosis prediction.

### 4.6. Limitations of Data Available

This review highlights the limited primary literature that is currently available for the association of PGs/GAGs with TAAD. It also reveals that much available data is somewhat contradictory, suggesting that different PGs/GAGs likely have different roles in disease pathogenesis that require further exploration. Studies included in this review have countries of origin in the USA, Asia or Europe. However, incidence rates, treatment opportunities for TAAD and lifestyle differ by geographical location. Therefore, some of the discrepancies observed between studies may relate to geographical differences. Additionally, many of the studies comparing TAAD with controls involve small sample sizes, with the majority of studies involving less than 10 patients per group, with an additional 5 studies having 10–20 participants per group and only 2 studies having over 20 patients per group; 27 patients in each group comparing control and TAD [79] and 30 control and 60 TAD patients [45] (Table 5). One study included different numbers of patients per group depending on analyses conducted, ranging from 6 per group to 33 TAD patients [43].

Discrepancies in PG/GAG reporting may also be due to development in techniques resulting in improvements in detection sensitivity, sub-typing capabilities and quantification that have occurred over the past 60 years. These include advances in mass spectrometry techniques, generation of improved antibodies and immunofluorescence methods. Since the 1960s, there has been a large change in the number and type of techniques used for GAG/PG isolation, identification and quantification.

A limitation of this type of review can occur due to the selection of search terms. As proteomics is a commonly employed technique to identify changes in protein expression levels between patient cohorts, additional studies of PG/GAG association with TAAD were also identified using the search terms proteomics, human and thoracic aneurysm/dissection. These studies highlighted changes in expression levels of PGs or associated proteins when comparing TAA or TAD tissue with neighbouring non-diseased or control tissue. Increased expression of perlecan and a cDNA clone with high similarity to biglycan and a decreased expression of chondroitin sulphate proteoglycan 4 was found in TAD patients [80]. In TAD patients with hypertension decorin, versican core protein and basement membrane-specific heparan sulphate proteoglycan core protein precursor were downregulated [81]. This study also suggested that decorin acts via the TGF-β signalling pathway to alter ECM composition. In contrast to studies reported in Table 3, proteomics analysis identified decreased expression of basement membrane-specific heparan sulphate proteoglycan core protein, mimecan, versican and aggrecan core proteins and hyaluronan and proteoglycan link protein 1 and decreased decorin in TAA compared to adjacent non-aneurysmal tissue [82,83]. Additionally, lumican showed decreased expression in medium-sized (5–5.5 cm) aneurysms compared with control patients [84]. Analysis of calcified TAAs identified decorin, lumican and biglycan as increased, with differential effects in mimecan (increased in 3, decreased in 3 out 6), aggrecan core protein (decreased in 4 out of 6, no change in 1 and increased in 1), versican core protein (decreased in 5 out of 6, increased in 1 out of 6) [85]. These studies further contribute to the contradictory findings on the potential association of PGs in TAAD. Some of the highlighted proteomics studies focus on specific sub-populations within TAAD, e.g., calcified regions or hypertension, so findings may be related to pathways associated with these potential confounding factors. These studies further highlight the need for larger, more detailed investigations on PG/GAG presence and potential role in TAAD and sub-populations within these disease cohorts.

**Table 5 ijms-23-09200-t005:** Patient numbers in TAA and TAD studies.

Study	Normal	TAA	TAD
[40]	10		10
[48]			14 (acute)
			3 (chronic)
[42]	3	3	
[53]	7	8	
[52]	19		16
[45]	30		60
[47]	15		21 (acute)
			8 (chronic)
[27]	9		10
[78]			58
[43]	12	13	33
	7		14
	6		6
[79]	27		27
[41]	3		3
[51]	12	14	16
[50]	8	10	10
[86]	10	9	
[46]	13		13

This review has focused on sporadic TAAD; however, knowledge from syndromic patients or animal models generated to investigate syndromic conditions can also contribute to the understanding of the role of PGs in TAAD. TGF-β mutations are well established to contribute to aortic pathologies. Versican and hyaluronan expression are induced by TGF-β, with increased hyaluronan found in Marfan syndrome patients [87]. Animal models of Marfan syndrome with mutations that reduce the amount of fibrillin-1 produced to display a reduction in cleaved aggrecan compared with wild-type mice [42]. ADAMTS-5, a protease associated with aggrecan and versican degradation, displays reduced expression in TAD patients [42]. This highlights the role that proteolytic cleavage likely contributes to increased PG levels (specifically aggrecan) observed in TAAD. A complete analysis of RNA and protein levels together with degradation pathways is required to determine the underlying mechanisms associated with altered PG/GAG levels (Figure 4). Much of our understanding of the role of PGs and associated genes to date have come from associated syndromic conditions or animal models. Human TAAD tissue that is available for research and composition assessment is obtained during surgery or the following post-mortem and, therefore, represents an end-point for disease, meaning that establishing PGs/GAGs as initiating factors or the association with other stimuli during disease processes is difficult/impossible. Given this lack of ability to study tissue composition during disease initiation and progression in humans, animal studies may represent the best way to probe cause and effect relationships.

## 5. Conclusions: Is There Enough Evidence to Suggest If GAGs/PGs Have a Role in TAAD?

PGs have a range of physiological functions, including roles in cell proliferation, adhesion, signalling, maintaining ECM homeostasis and local pressure. They function as part of a cycle where local tissue changes would affect PG levels and activity, with a corresponding alteration in PG/GAGs causing adaptation of local tissue properties (Figure 4). It, therefore, seems likely that GAGs/PGs are involved in TAA and TAD; however, it is not currently clear whether they have a role in the initiation of disease or whether they play a role in response to early disease-associated changes, e.g., in wound healing following initial dissection event. Many of the studies included in this review have small study numbers which may contribute to the current contradictory data available. More studies using larger participant numbers are needed to clarify the association with disease. However, as described previously, this is difficult to do using human tissue samples; therefore, animal studies or the use of known syndromic conditions may play essential roles in this understanding. This review has highlighted several avenues associated with PG/GAGs that may provide an opportunity for enhancing understanding of disease initiation or progression or towards therapeutic targeting approaches.

## 6. Limitations of the Study

Search terms used were designed to encompass as many papers as possible; however, some relevant studies without these specific terms may have been missed. Additionally, paid-for papers that were not under the subscription of the University of Liverpool were excluded, meaning key studies may not be included. Due to variability in study designs, quality assessment and comparisons across different studies were not possible.

## Figures and Tables

**Figure 1 ijms-23-09200-f001:**
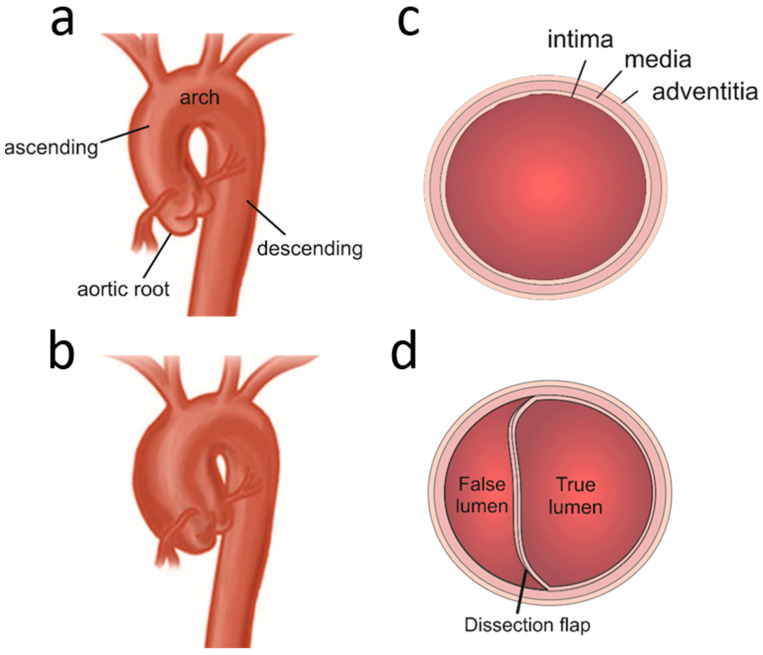
**Thoracic Aortic Aneurysm and Dissection.** (**a**) Normal thoracic aorta displaying the root, arch and descending thoracic aorta. (**b**) Depiction of an aneurysm in the ascending thoracic aorta showing dilatation of the aortic wall. (**c**) Cross-sectional image of the aortic wall showing the three layers of the wall; intima, media and adventitia. (**d**) Cross-sectional image of an aortic dissection represented by an intimal tear leading to accumulation of blood in the subintimal space, gradually expanding the false lumen created by the tear and generating a dissection flap.

**Figure 2 ijms-23-09200-f002:**
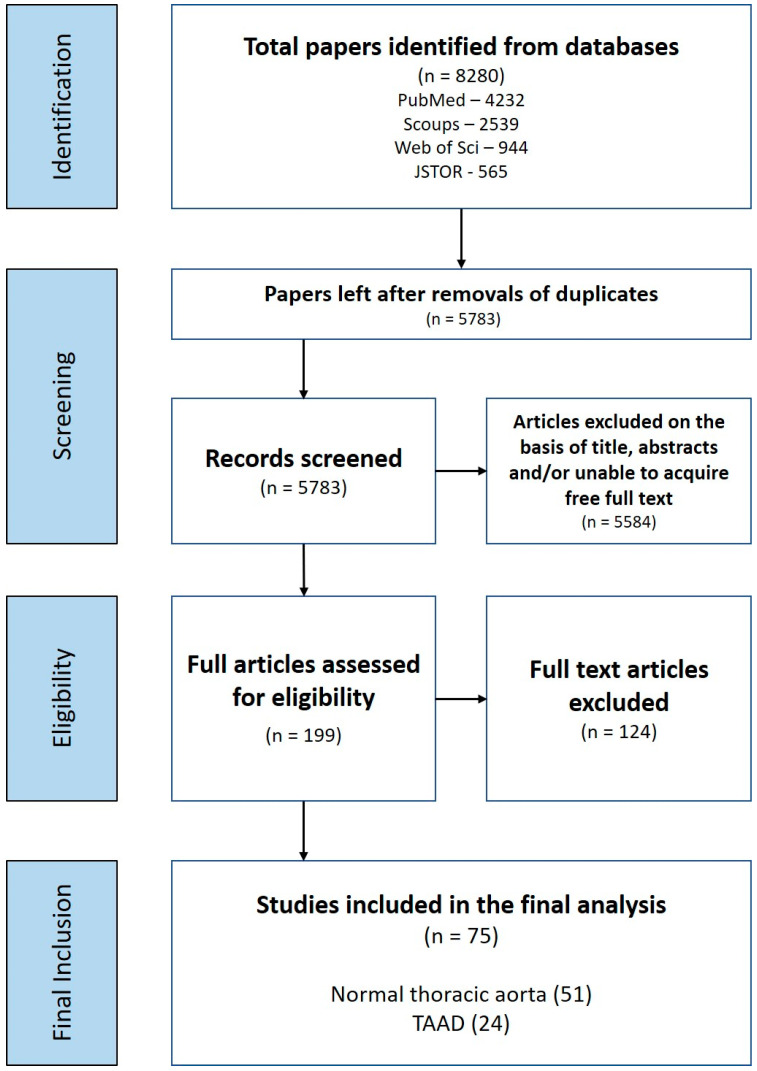
**Search process for healthy aorta and sporadic TAAD**. PRISMA flow depicting the search process for the role of GAGs/PGs in the normal aorta and sporadic TAAD. All papers were included as a whole throughout the data extraction, then separated into two categories; normal thoracic aorta (51) and TAAD (24), after the final inclusion stage.

**Figure 3 ijms-23-09200-f003:**
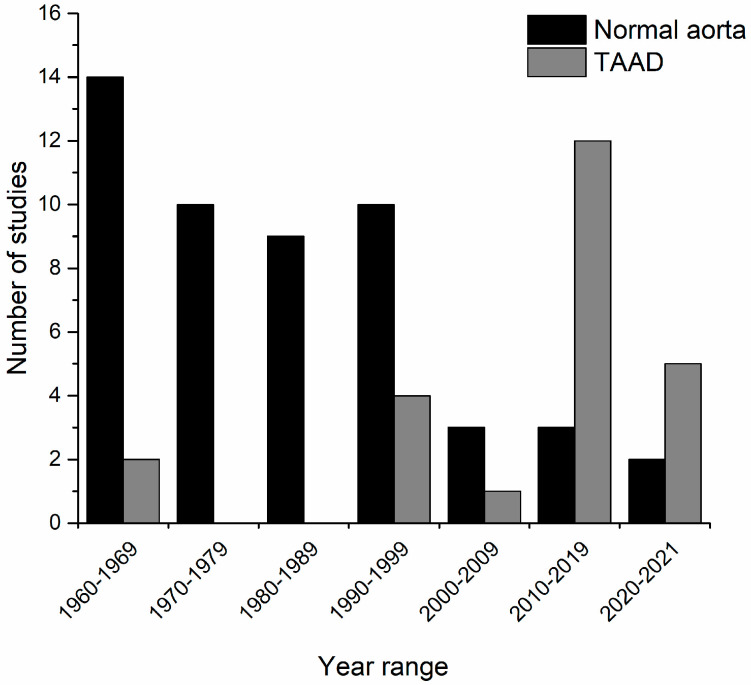
**Number of studies in each category included in this review sub-divided by decade.** Early studies largely focused on technique optimisation using normal aortic tissue, with TAAD studies following.

**Figure 4 ijms-23-09200-f004:**
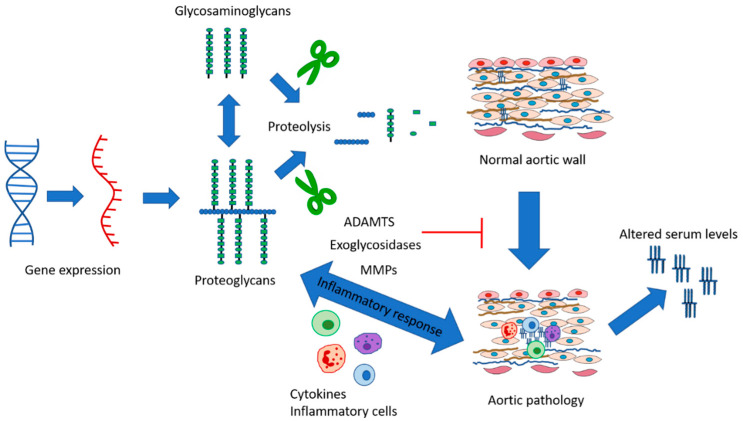
**Schematic showing the role that PGs/GAGs play in maintaining aortic wall integrity and association with aortic pathology.** Several points on this pathway have been shown to have altered levels in TAAD compared to the normal aorta, including RNA expression, PG/GAG and proteolysis enzymes with resulting alterations to ECM homeostasis, in turn altering wall structure and contributing to or initiating aortic pathologies. Altered serum levels of PGs have also been reported to be associated with aortic pathologies. These steps highlight areas for further research to understand the roles of individual components better and also provide avenues for future exploration as biomarkers or therapeutic targeting approaches.

**Table 1 ijms-23-09200-t001:** **Inclusion and exclusion criteria for healthy aorta and TAAD.** The criteria were defined before conducting the search for article selection for both normal and TAAD studies.

Inclusion Criteria	Exclusion Criteria
Free full text available (articles from journals under the subscription of the University of Liverpool)	Any papers that required payment, conference papers, abstract only, case report studies, commentaries, reviews, chapters and letters
Studies conducted on human thoracic aorta	Abdominal aorta, animal or cell-only studies
GAGs/PGs reported in association with normal thoracic aorta or sporadic TAAD	Any other associations of GAGs/PGs, e.g., in diabetes, hyperlipidaemia, etc., studies on mucopolysaccharidosis, syndromic TAAD
English Language only	Languages other than in English

**Table 4 ijms-23-09200-t004:** **Animal studies associated with PG/GAG changes.** Model, observations of altered PGs/GAGs and their effect on aortic pathology and integrity are summarised. Smooth muscle cells (SMC), angiotension II (AngII) β-aminopropionitrile (BAPN).

Study	Animal Model	PG/GAG Altered	Effect
[71]	ADAMTS5 deficient	Increased aggrecan/reduced cleavage	Aortic wall defects (elastin degradation, SMC loss)
[72]	ADAMTS5 deficient and AngII	Increased versican/reduced cleavage Increased ADAMTS1	Aortic dilatation
[42]	Marfan’s	Increased aggrecan and versican Reduced ADAMTS5	Dissection and rupture
[73]	ADAMTS9 deficient	Increased versican/reduced cleavage	Aortic wall defects
[50]	ADAMTS4 deficient and AngII	Reduced versican degradation	Reduced aortic wall defects (elastic fibre destruction, inflammation and SMC apoptosis) Reduced aneurysm, dissection and rupture
[52]	AngII	Increased versican degradation, increased ADAMTS1 expression	Dissection
[48]	Lumican deficient, AngII and BAPN		Increase in aortic rupture and dissection-associated mortality
	AngII and BAPN	Increased lumican levels	
[74]	Biglycan deficient and BAPN	Increased vascular perlecan	Aneurysm and rupture Aortic wall defects (disrupted collagen and elastin)
[70]	Biglycan deficient		Dissection and rupture Structural abnormalities and altered tensile strength of collagen
[75]	AngII		Remodelling of the aortic wall (SMC synthesis of new collagen colocalised with increased GAG production)
[46]	BAPN	Reduced decorin	Aortic wall defects (elastin)
	BAPN with Sirtuin 1 activator	Partial restoration of decorin levels	Protects against BAPN-induced aortic wall defects (elastin)
[76]	Contractile protein mutations with hypertension	GAG pools observed in delaminated vessels	Mortality Aortic wall defects (delamination)
[77]	Perlecan deficient		Dissection Aortic wall defects (thin/torn elastic lamina, immature elastic fibres)

## Data Availability

Data extraction tables available upon request.

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
