# Peer review of "Is There Enough Evidence to Support the Role of Glycosaminoglycans and Proteoglycans in Thoracic Aortic Aneurysm and Dissection?—A Systematic Review"

_ijms, 2022, doi:10.3390/ijms23169200_

Round 1

Reviewer 1 Report

Is there enough evidence to support the role of glycosaminoglycans and proteoglycans in thoracic aortic aneurysm and dissection? – a systematic review

While this review is an interesting concept, I have difficulty with the selection criteria, whereby the authors have missed a large number of manuscripts where TAA tissue has been studied and glycoprotein data has been found. That is my main concern. When using search terms proteomics AND human AND thoracic aneurysm approx. 40 papers appeared of which the below ones seem relevant and not cited in the review. Because of the rigid structure of a systematic review, which is unusual in a basic research setup exactly because of the problem of missing data, it will probably be hard to incorporate all these manuscripts in the current analyses. But then at least discuss all the relevant manuscripts where the different glycoproteins/PG or GAG have been found in the data or supplemental data of these manuscripts. It is a true limitation of the current setup that the search was not performed also on the term proteomics. I wonder if the analyses of the data becomes more clear upon addition of these manuscripts.

Aging Alters the Aortic Proteome in Health and Thoracic Aortic Aneurysm.

Tyrrell DJ, et al. Arterioscler Thromb Vasc Biol. 2022 May 5:101161ATVBAHA122317643. doi: 10.1161/ATVBAHA.122.317643.

Multi-Omics Profiling in Marfan Syndrome: Further Insights into the Molecular Mechanisms Involved in Aortic Disease.

Verhagen JMA, et al. Int J Mol Sci. 2021 Dec 31;23(1):438. doi: 10.3390/ijms23010438

Aorta smooth muscle-on-a-chip reveals impaired mitochondrial dynamics as a therapeutic target for aortic aneurysm in bicuspid aortic valve disease.

Abudupataer M, et al. Elife. 2021 Sep 6;10:e69310. doi: 10.7554/eLife.69310.

New mechanistic insights to PLOD1-mediated human vascular disease.

Koenig SN, et al. Transl Res. 2022 Jan;239:1-17. doi: 10.1016/j.trsl.2021.08.002.

Evaluation of the Aortopathy in the Ascending Aorta: The Novelty of Using Matrix-Assisted Laser Desorption/Ionization Imaging.

Mohamed SA, et al. Proteomics Clin Appl. 2021 Jan;15(1):e2000047. doi: 10.1002/prca.202000047.

Quantitative proteomics reveal lineage-specific protein profiles in iPSC-derived Marfan syndrome smooth muscle cells.

Iosef C, et al. Sci Rep. 2020 Nov 23;10(1):20392. doi: 10.1038/s41598-020-77274-w.

TMT-Based Quantitative Proteomic Analysis Identification of Integrin Alpha 3 and Integrin Alpha 5 as Novel Biomarkers in Pathogenesis of Acute Aortic Dissection.

Xing L, et al.  Biomed Res Int. 2020 Jul 2;2020:1068402. doi: 10.1155/2020/1068402.

Glycoproteomic Analysis of the Aortic Extracellular Matrix in Marfan Patients.

Yin 殷晓科 X, et al. Arterioscler Thromb Vasc Biol. 2019 Sep;39(9):1859-1873. doi: 10.1161/ATVBAHA.118.312175.

Proteomic study of the microdissected aortic media in human thoracic aortic aneurysms.

Serhatli M, et al. J Proteome Res. 2014 Nov 7;13(11):5071-80. doi: 10.1021/pr5006586.

Angiogenesis and remodelling in human thoracic aortic aneurysms.

Kessler K, et al. Cardiovasc Res. 2014 Oct 1;104(1):147-59. doi: 10.1093/cvr/cvu196.

Comparative tissue proteomics analysis of thoracic aortic dissection with hypertension using the iTRAQ technique.

Zhang K, et al. Eur J Cardiothorac Surg. 2015 Mar;47(3):431-8. doi: 10.1093/ejcts/ezu171.

Proteomic comparison between abdominal and thoracic aortic aneurysms.

Matsumoto K, et al. Int J Mol Med. 2014 Apr;33(4):1035-47. doi: 10.3892/ijmm.2014.1627.

Preliminary biomarkers for identification of human ascending thoracic aortic aneurysm.

Black KM, et al. J Am Heart Assoc. 2013 Nov 14;2(6):e000138. doi: 10.1161/JAHA.113.000138

A combined proteomic and transcriptomic approach shows diverging molecular mechanisms in thoracic aortic aneurysm development in patients with tricuspid- and bicuspid aortic valve.

Kjellqvist S, et al. Mol Cell Proteomics. 2013 Feb;12(2):407-25. doi: 10.1074/mcp.M112.021873.

EcmPred: prediction of extracellular matrix proteins based on random forest with maximum relevance minimum redundancy feature selection.

Kandaswamy KK, et al. J Theor Biol. 2013 Jan 21;317:377-83. doi: 10.1016/j.jtbi.2012.10.015.

Proteomic analysis of calcified abdominal and thoracic aortic aneurysms.

Matsumoto K, et al. Int J Mol Med. 2012 Aug;30(2):417-29. doi: 10.3892/ijmm.2012.985.

Proteomic analysis in aortic media of patients with Marfan syndrome reveals increased activity of calpain 2 in aortic aneurysms.

Pilop C, et al. Circulation. 2009 Sep 15;120(11):983-91. doi: 10.1161/CIRCULATIONAHA.108.843516.

Author Response

We have carried out additional searches as suggested by the reviewer and identified manuscripts out of the list provided and others from the searches that report changes in PGs relevant to TAAD. Discussion of these additional studies have been included in a new paragraph in the discussion.

Author Response

We have amended these overlapping statements.

Reviewer 3 Report

This article includes a very complete revision of the possible role of GAGs/PGs in aortic disease. I have some minor comments for the authors:

-Table 2 provides very useful information and sums up all the results from all the papers mentioned in the text. I would add also the technique that the authors used in those articles to measure GAG levels. It is not mentioned in the text and it would be helpful for the readers to have that information.

-I would add a Table summarizing  results in animals models.

Author Response

We have added methods used into Table 2 and added a Table summarising animal model data as suggested by the reviewer.